# OpenReview forum: "LLMScan: Causal Scan for LLM Misbehavior Detection"
_ICML.cc/2025/Conference — ICML 2025 poster_

### Official Review · Reviewer_XgXg · 2025-03-07

**Overall Recommendation:** 4

**Summary:**

The authors propose to detect misbehavior in LLMs through two major components. 1) They assess the contribution of individual input tokens and neural network layers on the final output. 2) They train a detector to classify prompts based on the properties of the analysis conducted in 1). They evaluate their approach on truthfulness, harmfulness, and toxicity and conduct experiments on 56 benchmarks.

References for the remaining review:

[1] Carlini et. al., "Adversarial Examples Are Not Easily Detected: Bypassing Ten Detection Methods", 2017
[2] Schwinn et. al., "Soft Prompt Threats: Attacking Safety Alignment and Unlearning in Open-Source LLMs through the Embedding Space", 2024

## update after rebuttal

Most of my concerns have been addressed by the reviewer. I specifically liked the ablation study the authors provided based on the question of reviewer **4jWw** regarding individual components of their algorithm.
I would like to highlight the code should be released upon acceptance of this paper, which is the most major concern I currently have with the paper (no supplement).

I raised my score to "4"

**Claims And Evidence:**

The authors claim:
- To propose a novel method to detect misbehavior in LLMs
- That their method identifies misbehavior successfully for 4 different types on 56 benchmarks

The authors provide:
- Results on a large number of datasets

**Essential References Not Discussed:**

I am not aware of essential references that were not discussed

**Experimental Designs Or Analyses:**

The claims made by the authors are well substantiated by the experiments performed. Moreover, the authors perform additional ablation studies to investigate the working mechanisms and design choices of the proposed algorithm

**Methods And Evaluation Criteria:**

The authors use common evaluation benchmarks and compare their proposed approach with recently proposed methods, including highly cited works. Moreover, the authors provide code, which makes it easier to understand the exact implementation settings of the experiments. I was not able to spot inconsistencies in the evaluation.

**Other Comments Or Suggestions:**

- remove inconsistency in notation regarding definitions (e.g., using square symbol in the end)
- section three presents very simple concepts in an overly complicated way.
- Overall the figures and tables take a lot of space (Table 2 and Figure 4) present relatively simple ablations but take considerable space. Some of the results could be moved to the appendix while keeping the general message.

**Other Strengths And Weaknesses:**

Strengths:

- comprehensive benchmark including several baselines, multiple variations of misbehavior, and a large number of different datasets
- their method consistently outperforms baselines with few exceptions despite the variety of settings

Weaknesses:

- Adversarial example detection has shown to be considerably harder than estimated by several works [1]. The authors should highlight that their approach detects black-box adversaries not aware of the defense mechanism. A study with an adaptive attack trying to bypass the defense would be interesting. This could be achieved through continuous attack with reasonable effort (as a proof of concept) [2] (however, the proof of concept is a minor concern).

- The captions of tables and figures could be more comprehensive (minor).

- The fonts in FIgure 3 are very hard to read (minor).



I am currently between weak reject and weak accept and will reassess my decision based on the other reviews and the rebuttal of the authors.

**Questions For Authors:**

- Could the authors provide an estimate of the computational overhead of their algorithm compared to the baseline approaches?
- Could the authors give their opinion on the ability of a white-box adversary to bypass their detection method?

# After rebuttal:

Most of my concerns have been addressed by the reviewer. I specifically liked the ablation study the authors provided based on the question of reviewer **4jWw** regarding individual components of their algorithm.
I would like to highlight the code should be released upon acceptance of this paper, which is the most major concern I currently have with the paper (no supplement).

I raised my score to "4" for now. I will continue to follow the discussion and change my score accordingly if it should be required.

**Relation To Broader Scientific Literature:**

The authors provide a comprehensive overview concerning related works. I am not well familiar with some of the literature but was able to put the work into context due to the references provided by the authors.

**Theoretical Claims:**

N/A

---

> ### Author Rebuttal · Authors · 2025-04-01
>
> Thank you for taking the time to review our paper and for your insightful comments. Please find our responses to your questions below.
>
> **Q1.** Computational overhead compared to the baseline approaches.
>
> **A1.** We thank the reviewer for highlighting the importance of evaluating the computational overhead of our method. To this end, we conducted experiments comparing the average detection time per input with baseline methods. We randomly sampled 100 prompts (average length: 30 tokens) and measured detection time across the benchmark models. The results show that our method introduces moderate overhead, remaining within the same order of magnitude as the model’s inference time. The table below summarizes the detection time (in seconds) and the corresponding inference time for reference.
>
> |Model|Llama-2-7b|Llama-2-13b|Llama-3.1|Mistral|
> |-|-:|-:|-:|-:|
> |LLMScan|3.82|7.73|3.31|3.40|
> |Lie Detector|6.79|13.24|7.53|8.24|
> |RepE|0.05|0.08|0.05|0.05|
> |ONION|0.30|0.30|0.30|0.30|
> |Complete inference time|3.07|6.68|3.37|3.54|
>
> On average, for 7B–8B models (Llama-2-7B, Llama-3.1, Mistral), our method requires around 3.5 seconds per input, compared to 7.5 seconds for the baseline lie detector and 0.05 seconds for RepE. For Llama-2-13B, it takes 7.7 seconds, while the lie detector baseline requires 13.2 seconds and RepE 0.08 seconds. For backdoor detection, the ONION baseline works as a pre-processing step and takes approximately 0.3 seconds per input across all models.
>
> While RepE is the fastest due to its single-pass extraction, it sacrifices detection reliability, as discussed in the paper. Similarly, ONION performs poorly in backdoor detection. In contrast, our method is more efficient than the baseline lie detector while achieving stronger detection performance.
>
> In summary, our method introduces moderate overhead and strikes a balance between efficiency and effectiveness. We have included a detailed breakdown of runtime performance across model sizes in the appendix.
>
> **Q2.** Could the authors give their opinion on the ability of a white-box adversary to bypass their detection method?
>
> **A2.** We thank the reviewer for raising this important point regarding the robustness of our method against white-box adversaries. We acknowledge that adversarial detection is particularly challenging under adaptive attacks and appreciate the opportunity to clarify the threat model in our work. While a white-box adversary could theoretically attempt to bypass our detection by minimizing causal signals, such attacks are highly non-trivial in practice for two reasons:
>
> Layer-level causal effects in our method are discrete values based on separate interventions. This process produces non-differentiable outputs and discrete shifts in behavior. As a result, the causal signals they generate are not amenable to standard gradient-based optimization techniques. This makes it challenging even for a white-box adversary to perform targeted manipulation.
>
> Token-level causal effects are based on statistical aggregation across calculated distances between intervened attention heads and the original one. This complex calculation process makes them inherently noisy and non-smooth. As a result, an adversarial would need to account for a wide range of token-level variations, as well as attention heads changes. This would significantly complicate the optimization process potentially adopted by an adaptive attacker.
>
> We have added a discussion to clarify that our experiments were conducted to detect adversaries who are not aware of the defense mechanism, and a discussion on the difficulty of conducting adaptive attack.
>
> **Comments:**
>
> **C1.**: remove inconsistency in notation regarding definitions.
>
> **A1.** Thank you for pointing out the inconsistency in our submission, and we have revised accordingly
>
> **C2.** section three presents very simple concepts in an overly complicated way.
>
> **A2.** We thank the reviewer for the comment. Our intention was to provide a formal foundation for our causal analysis, while acknowledging the practical compromises needed for token and layer level analysis. We acknowledge the reviewer’s concern that parts of Section 3 may have introduced unnecessary complexity and redundancy. In response, we have streamlined the definitions and moved some of them to the appendix.
>
> **C3.** Overall the figures and tables take a lot of space.
>
> **A3.** We will revisit the figures/tables and move them to appendix in case space is required (to clarify all issues during the rebuttal).
>
> **Other response:**
> We acknowledge the readability issue in Figure 3 and have revised it with larger fonts for better visibility. Additionally, we have expanded the captions of tables and figures to make them more informative. For example, the caption of Figure 2 now clarifies that it illustrates combined causal effects at the prompt and layer levels for a truth and lie response.

---

### Official Review · Reviewer_RtXP · 2025-03-11

**Overall Recommendation:** 4

**Summary:**

The authors present LLMScan, a technique to determine when the LLM is misbehaving via causal inference. They analyze the causal effects of input tokens by performing interventions on each token and measuring changes in attention scores, and in transformer layer by skipping the layer and comparing the output logits. From these changes, they create a casual effect input which is fed to a classifier which predicting if the model is misbehaving or not.

**Claims And Evidence:**

Weaknesses
- The authors mention generating a causal map but from Figure 2 they show a single list of values for both the input tokens and layers.

- While their approach shows encouraging results, the paper lacks a rigorous mathematical framework to call this a “causal effect”, as when we skip a layer or change an input token, it will have other downstream effect on the output. Additionally adding instructions like "Answer the following question with a lie." can drastically change the output.

**Essential References Not Discussed:**

Not sure

**Experimental Designs Or Analyses:**

Yes

**Methods And Evaluation Criteria:**

Yes

**Other Comments Or Suggestions:**

No

**Other Strengths And Weaknesses:**

Strengths:
- This approach can detect different types of LLM misbehavior
- The performance is pretty good specially for Backdoor detection

**Questions For Authors:**

No

**Relation To Broader Scientific Literature:**

Analyzing changes in attention maps with changes per token for detecting misbehaviors is useful.

**Theoretical Claims:**

NA

---

> ### Author Rebuttal · Authors · 2025-04-01
>
> Thank you for taking the time to review our paper and for your insightful comments. Please find our responses to your questions below.
>
> **Q1.** Causal map visualization showing both input tokens and layers.
>
> **A1.** We thank the reviewer for their careful observation regarding the presentation of the causal map in Figure 2 and sorry for the confusion. In our work, the term “causal map” refers to the combined causal effects computed across two dimensions including 1) token-level causal effects, which correspond to contribution of the input (embedding); 2) layer-level causal effects, which correspond to contribution of the transformer layers. We chose the term as it resembles a “heat map” over the input and the internal layers.
>
> **Q2.** The “causal effect” definition and concerns about downstream effects after intervention.
>
> **A2.** We appreciate the reviewer for raising this thoughtful concern regarding the formal definition of causal effect in our work. As discussed in Section 3.1, a proper causality analysis would require a formal model such as structured causal model, and systematic intervention (as shown in equation 3). Such analysis is, however, often impractical as it is computationally intractable. What we adopted in this work is a lightweight and practical approach that is referred to in prior works [1, 2] as causal mediation analysis (CMA). In CMA, causal effects are approximated by comparing outcomes between the normal execution and an abnormal execution. In our context, the normal execution refers to the LLM’s standard output and the abnormal execution corresponds to the output when we apply targeted interventions, i.e., modifying a specific token or skipping a transformer layer. The causal effect is then approximated by measuring the difference between the two executions. We acknowledge the reviewer’s concern that interventions like changing a token or skipping a layer may introduce cascading effects throughout the model. Nonetheless, we show that such poor approximation of the actual causal effect is already effective for detecting LLM misbehavior. We have added more clarification on our causality analysis method and its limitation in the revised submission to clarify this issue.
>
> We also appreciate the insightful comments on the downstream impact of adding instruction. This is in fact a question on the definition of misbehavior. Indeed, our definition of whether a model misbehaves (e.g., whether it is generating toxic responses) does not depend on the prompt. In other words, our method will detect misbehavior itself (such as generating toxic responses) even if the model is instructed to do so (such as with the instruction “answer the following question with a lie”). Such a choice could be considered reasonable if we aim to detect jailbreak attacks such as DAN that aims to induce toxic responses with “legit” excuses. Having said that, we agree that further research on detecting misbehavior considering the context is an extremely interesting one.
>
> Reference:
>
> [1] Locating and Editing Factual Associations in GPT (NeurIPS 2022)
>
> [2] Investigating Gender Bias in Language Models Using Causal Mediation Analysis (NeurIPS 2020)

---

> > ### Comment · Reviewer_RtXP · 2025-04-07
> >
> > Thanks for addressing the issues I have. I have raised my score to 4.

---

### Official Review · Reviewer_4jWw · 2025-03-12

**Overall Recommendation:** 3

**Summary:**

In this paper, the authors introduce a novel framework for detecting misbehavior in LLMs using causal analysis. In particular, the proposed framework consists of two main modules: i) a scanner that can conduct causality analysis at both token and model layer levels, and ii) a detector which is trained on causal maps to identify LLM misbehaviors based on the generated causality distribution maps.

**Claims And Evidence:**

The authors conduct intensive experiments to evaluate the proposed approach. However, more evidence is needed as “empirical results” supported the configuration of attention head distances and layer selection. It would be better to demonstrate and analyze the empirical results.

**Essential References Not Discussed:**

The authors introduced related work on the four different misbehavior detection scenarios. I suggest that the authors can consider further clarifying the research gap in Section 2.

**Experimental Designs Or Analyses:**

The authors employ Llama family and Mistral to evaluate the proposed method regarding AUC and accuracy, and also perform ablation study to assess token-level and layer-level causal effects.
- Table 2: toxicity detection, layer level, why 0.98 for Mistral is not in bold? The result is better than the other three LLMs at the layer level.

**Methods And Evaluation Criteria:**

The proposed method is evaluated regarding accuracy and AUC score via comparative analysis on 14 datasets, and ablation study on the contribution of token-level and layer-level causal effects.
- In Section 3.1, the authors mention that for lie detection, the LLM “is induced to lie with the instruction”. In this case, does the LLM still misbehave? Same question for the toxic instruction prompts.

**Other Comments Or Suggestions:**

- I wonder what the application scenario is, real-time detection or retrospective analysis?
- Section 1: “To evaluate the effectiveness of LLMSCAN, we conduct experiments using four popular LLMs across 13 diverse datasets.” I assume it should be 14 datasets based on the evaluation section.
- Section 4.2: “This suggests that truthful responses concentrate relevant knowledge in select layers …” -> “selected”.

**Other Strengths And Weaknesses:**

The paper is well-organized, and the language is technical yet understandable for readers with domain knowledge. The figures are clear and the overall readability is good.

**Questions For Authors:**

- In Section 3.1, the authors mention that for lie detection, the LLM “is induced to lie with the instruction”. In this case, does the LLM still misbehave? Same question for the toxic instruction prompts.
- How do you obtain the “empirical results” for attention head distances and layer selection?
- Table 2: toxicity detection, layer level, why 0.98 for Mistral is not in bold? The result is better than the other three LLMs at the layer level.
- I wonder what the application scenario is, real-time detection or retrospective analysis?

**Relation To Broader Scientific Literature:**

The authors provide impact statement at the end of the manuscript.

**Theoretical Claims:**

The authors propose three definitions for generative LLM, token-level and layer-level causal effect respectively.

---

> ### Author Rebuttal · Authors · 2025-04-01
>
> Thank you for taking the time to review our paper and for your insightful comments. Please find our responses to your questions below.
>
> **Q1.** Concerns on LLM is induced to misbehavior (e.g., lie) with instruction.
>
> **A1.** We thank the reviewer for the careful evaluation and for raising an important question about the definition of misbehavior. In our work, we consider the model to misbehave regardless of whether the behavior is prompted. That is, our detection focuses on the output behavior itself (e.g., generating toxic content), not the intent of the prompt. This design aligns with real-world scenarios like jailbreak attacks (e.g., DAN) that induce harmful outputs through seemingly legitimate instructions. Having said that, we agree that further research on detecting misbehavior considering the context is an extremely interesting one.
>
> **Q2.** “Empirical results” to support the configuration.
>
> **A2.** We thank the reviewer for highlighting the need for more empirical evidence to support our design choices and appreciate the opportunity to clarify how these configurations were determined. In response, we have conducted additional experiments on each task and included the results under the appendix in the revised submission.
>
> Our design choices were motivated by both prior research and the need to balance detection accuracy and efficiency. Specifically, we chose to use attention head outputs instead of hidden states for two main reasons. First, prior studies [1, 2] show that attention layers effectively encapsulate the model’s stored knowledge, including harmful or inappropriate content—making attention heads a more direct and sensitive signal for detecting misbehavior. Second, attention head outputs are considerably smaller than hidden states, reducing computational overhead. To further balance detection accuracy and efficiency, we adopt a sparse sampling strategy. For layer selection, we sample three representative layers from the early, middle, and late stages of the model, reflecting the progressive evolution of internal representations [3]. This also loosely mirrors how humans process information, from perception to reasoning. For each selected layer, we sample three attention heads to ensure functional diversity without incurring unnecessary overhead. We have clarified these design choices in the revised submission.
>
> In our empirical evaluation, we compared our method against two baselines to assess the effectiveness of our sparse sampling strategy. The first baseline applies LLMScan with all layers, sampling three attention heads per layer, following our standard head selection. The second baseline uses all attention heads but only from the three selected layers. We conducted experiments on four detection tasks, each with one representative dataset. The results, shown below (token-level detection accuracy), demonstrate that our sampling strategy achieves comparable or better performance while avoiding unnecessary noise:
>
> |Task|Dataset|LLMScan (Ours)|LLMScan w/ all layers|LLMScan w/ all attention heads|
> |-|-:|-:|-:|-:|
> |Lie Detection|Question1000|0.94|0.88|0.93|
> |Jailbreak Detection|AutoDAN|0.97|0.95|0.96|
> |Toxicity Detection|SocialChem|0.61|0.59|0.62|
> |Backdoor Detection|CTBA|0.96|0.94|0.92|
>
> This confirms that selecting representative layers and heads is sufficient to capture the key causal signals for effective detection.
>
> Reference:
>
> [1] Locating and Editing Factual Associations in GPT (NeurIPS 2022)
>
> [2] On the Role of Attention Heads in Large Language Model Safety (ICLR 2025)
>
> [3] Transformer Feed-Forward Layers Are Key-Value Memories (EMNLP 2021)
>
> **Q3.**  About the understanding of Table 2: toxicity detection.
>
> **A3.** We thank the reviewer for their careful reading and for pointing out the potential confusion regarding the bold values in Table 2. At table 2, we compare the performance of our method based on token-level causal effect only and layer-level causal effect only, on the same task and the same models. We highlighted the better results. For example, for toxicity detection tasks, the token-level performance for the Mistral model is 1.0 accuracy. Since the accuracy of the layer-level analysis is only 0.98, we highlighted the token-level performance for this case. We have revised the table caption to clarify this.
>
> **Q4.** Application scenario: real-time detection or retrospective analysis?
>
> **A4.** We thank the reviewer for raising this question regarding the application scenario. Our method is primarily designed for real-time detection, as it analyzes the LLM’s internal behavior during inference to enable on-the-fly detection of potential misbehavior. We have added a detailed discussion on the intended usage and the computational overhead of runtime detection in the appendix (see our replies to Reviewer 8BQP-Question 2, and Reviewer RtXP-Question 1 for details).
>
> **Others**
>
> Thanks for pointing out the typos. We fixed all of them in our revised submission.

---

> > ### Comment · Reviewer_4jWw · 2025-04-03
> >
> > I thank the authors for clarifying the questions and addressing my previous comments.

---

### Official Review · Reviewer_8BQP · 2025-03-13

**Overall Recommendation:** 4

**Summary:**

This paper proposes a detection mechanism named "LLMSCAN" for identifying potential "undesirable" generation behaviors during large language model (LLM) inference. The core approach is to utilize causal analysis to "intervene" in the input tokens and the transformer layers of the model, and to measure the causal impact values of each part based on the changes in the output distribution and attention. This forms a causal map. Then, based on these causal maps, the paper trains a classifier (detector) to judge whether the model may produce untruthful, harmful, toxic, or backdoor attack-induced abnormal outputs. The paper conducts experiments on multiple common LLMs and a wide range of benchmark tests (including lie detection, jailbreak attack detection, toxicity detection, backdoor attack detection, etc.), and the results show that this method has a high detection rate and accuracy, and can issue early warnings to avoid the generation of truly harmful content.

**Claims And Evidence:**

The paper claims that "LLMSCAN" can effectively distinguish between normal and malicious behavior by monitoring the intervention changes in the internal attention distribution and the skip outputs of some transformer layers. Overall, the experiments in the paper, compared with the baseline methods (such as black-box methods or traditional filtering methods based on output), all show excellent performance. The detection of various types of malicious behavior in this paper has given AUCs over 0.95 or equivalent high accuracy rates.

From the results given in the paper, most of these claims are supported by relatively strong data, each type of malicious behavior is compared with existing or classical detection methods, and it is confirmed that stable detection performance can be maintained under different model scales and different attack/error information scenarios. Since the experiments were conducted on multiple public benchmarks and provided ablation experiments (such as comparing detection performance using token-level causal features or layer-level causal features alone), this supports the authors' claims about the generalizability and effectiveness of the method. These pieces of evidence are consistent with the main theoretical concepts proposed in the paper.

**Essential References Not Discussed:**

I did not find the essential but omitted key references.

**Experimental Designs Or Analyses:**

I reviewed the paper's detection experiments on different LLMs (including Llama-2-7b, 13b, Mistral, etc.), comparison baselines, and tests on different types of malicious behaviors. The sampling, training-test split, and presentation of AUC and Accuracy were all clear, and the scale of the experiments was sufficient to demonstrate the effectiveness of the method. From the given experimental data, the experimental design is reasonable:

1. When comparing with similar methods (such as output-based detection or ONION defense, etc.), it showed a significant improvement in accuracy and AUC.
2. Similar regular conclusions were presented for models of different scales and different datasets.

These designs and analyses can well validate the proposed "LLMSCAN" method, and no major defects have been found so far.

**Methods And Evaluation Criteria:**

The author's method first performs two-level interventions on the LLM during reasoning: 1) Replace each input token with a special character one by one and observe the changes in attention scores; 2) Skip or "bypass" different transformer layers and compare the differences in output logits. Then, these difference information is combined to form a "causal map." A classifier is then used to distinguish these causal maps, with the goal of distinguishing normal outputs from different categories of abnormal outputs.

This design reasonably observes the internal attention mechanism and hierarchical outputs of the LLM for causal observations and uses them as feature inputs to a relatively lightweight detector. The evaluation criteria mainly rely on AUC, accuracy, and comparison with the baseline. These evaluation indicators are suitable for measuring detection tasks and cover the performance of methods under unbalanced or various scenarios.

**Other Comments Or Suggestions:**

The paper briefly mentions the impact on large models when skipping the specific implementation of a certain layer during the scanning process, such as whether it will bring pressure on memory or speed. Perhaps more technical details can be provided in the appendix, explaining the resource costs of executing this method in LLMs of different sizes, as well as whether it has an actual impact on the model's response speed.

**Other Strengths And Weaknesses:**

Advantages include the method's good versatility, as it can detect undesirable behaviors (lies, toxicity, backdoors, etc.) in different scenarios with a unified approach, and it is relatively efficient, requiring no significant modifications to the LLM itself. The theoretical part also closely adheres to the standard definition of causal analysis.

Disadvantages lie in the fact that the method requires inner access or modification to the LLM (such as hierarchical skipping and other interventions), which may not be feasible for commercial large models or API environments that are completely black-box. Additionally, in the experiments for backdoor attack and defense competitions, although the detection rate is high, some extreme prompts or attack methods may vary greatly in real-world scenarios. Relying on the accessibility of all model layers is a strong assumption.

**Questions For Authors:**

Have you considered the API access scenarios in actual deployment? Can this method simplify or approximate the implementation for commercial models that cannot directly access the output of the transformer layer?

**Relation To Broader Scientific Literature:**

The detection of adversarial examples, fake information detection, and the analysis of internal attention mechanisms are all important issues in the current natural language processing security field. Compared with methods that are only based on output text, LLMSCAN utilizes causal intervention mechanisms to evaluate the presence of potential harmful outputs from changes in internal activations and attention, which is consistent with the early exploration of hierarchical activations and attention in explainable AI or neural network visualization analysis. At the same time, the paper complements some previous works that mainly conduct security detection at the output or prompt level.

**Theoretical Claims:**

The paper does not contain large sections of pure theoretical derivation, but instead borrows the concept of causal inference to define "intervention" and "causal effect." The main formulas such as causal effect $CE_x$ or $CE_{x,\ell}$ and so on, all belong to the relatively standard "do operation" framework. Such definitions do not require very complex proof, and therefore there is no need for rigorous higher-order mathematical proof. As described, these theoretical parts are logically consistent and do not have obvious gaps. I have not found any serious mathematical errors that require refutation or correction.

---

> ### Author Rebuttal · Authors · 2025-04-01
>
> Thank you for taking the time to review our paper and for your insightful comments. Please find our responses to your questions below.
>
> **Q1.** API access scenarios.
>
> **A1.** We thank the reviewer for raising this important and practical question regarding the applicability of our method. Our method is primarily designed for users who train/finetune their own models and would like to monitor and diagnose their model when serving their customers. It assumes white-box access to the transformer layer outputs to compute token-level or layer-level causal effects. We do agree that this limits applicability to commercial black-box LLMs. This limitation can be potentially addressed through two ways:
>    1. Approximate token-level causal effect based on output distribution: In our method, we rely on logits different before and after intervention to compute the casual effect. If the logits are not available, we can replace it with token embedding differences or semantic difference before and after the intervention. Note that this wouldn't work for approximating the layer-wise causal effect.
>    2. Shadow model approximation: We would train a shadow model based on existing open-source LLMs and use the layer-wise casual effect of the shadow model to approximate the commercial one.
> We will explore both directions in future work and look forward to extending our method to broader settings when transformer internals are not accessible.
>
> **Q2.** Resource costs of executing this method in LLMs of different sizes.
>
> **A2.** We thank the reviewer for highlighting the importance of discussing the resource overhead of our method. To answer the question, we have run experiments to evaluate the overhead of our method.
>
> The efficiency of our method depends primarily on the input length and the number of models layers. Our experiments with an A100 server on the models studied in the submission show that layer-level causal effect computation takes about 0.08 seconds per layer, and token-level computation averages 0.04 seconds per token. Note that analyzing the casual effect of a layer or token takes much less time than generating the response since we only need to generate the first token to conduct the analysis.
>
> To evaluate the overall time overhead, we run the models with and without our method and compare the time. For models with 31 layers, such as Llama-2-7b, with our method, the models take around 6.89 seconds (3.82 seconds is spent on running our method) per input to generate the outputs, whereas without our method, the models take about 3.07 seconds to complete inference process. For models such as Llama-2-13b with 40 layers, it takes around 14.41 seconds (7.73 seconds is spent on running our method) per input with our method, and 6.68 seconds without our method. We remark that this overhead can be significantly reduced if we run our analysis and the original model in parallel, i.e., the overall time becomes 3.82 seconds and 7.73 seconds. The details of detection time are shown in the table below. Specifically, we randomly sampled 100 prompts with an average length of 30 tokens and measured the average detection time and inference time per input across all benchmark models.
>
> |Model|Llama-2-7b|Llama-2-13b|Llama-3.1|Mistral|
> |-|-:|-:|-:|-:|
> |LLMScan|3.82|7.73|3.31|3.40|
> |Complete inference time|3.07|6.68|3.37|3.54|
>
> In terms of memory overhead, if we run our method and the generation sequentially, there is little overhead; if we run them concurrently, because we need to create an additional model when analyzing the causal effect for each layer and each token (whilst the original model is executing to generate the response), the memory consumption doubles.
>
> We have added a new subsection to discuss the resource costs of our method under the appendix of our revised submission.

---

### Decision · Program_Chairs · 2025-05-01

**Decision:**

Accept (poster)

**Comment:**

The paper proposes a novel framework for detecting misbehavior in LLMs using causal analysis, which can effectively distinguish between normal and malicious behavior by monitoring the intervention changes in the internal attention distribution and the skip outputs of some transformer layers.

The proposed method is interesting and successfully identifies misbehavior for 4 different types on 56 benchmarks, demonstrating a significant improvement in accuracy and AUC compared with other baselines. All the reviewers lean to accept the paper. Therefore, I recommend to accept this paper.